# Understanding the Effect of Material Parameters on the Processability of Injection-Molded Thermoset-Based Bonded Magnets

**Uta Rösel *** and **Dietmar Drummer**

Institute of Polymer Technology, Friedrich-Alexander-Universität Erlangen-Nürnberg, 91058 Erlangen, Germany; dietmar.drummer@fau.de
* Correspondence: uta.ur.roesel@fau.de

**Abstract:** The applications of bonded magnets in the field of injection-molded samples can be expanded by thermoset-based polymer-bonded magnets, as thermosets provide the opportunity to comply with the demands of, for example, the chemical industry or pump systems in drive applications through to their improved chemical and thermal resistance, viscosity and creep behaviour, especially compared to thermoplastic-based magnets. This paper investigates the influence of the matrix material (epoxy resin, phenolic resin), the filler type (strontium-ferrite-oxide, neodymium-iron-boron) and the filler grade on the reaction kinetics and the viscosity. Based on the determination of the impact, the theory of the network structure is founded. The network and the cross-linked structure are essential to know, as they significantly define not only the material but also the sample behaviour. The correlation between the material system and the mechanical as well as the magnetic properties is portrayed based on the general understanding of the behaviour in terms of the reaction kinetics and the viscosity as well as the theory of the network structure. With that, a basic understanding of the correlation within the material system (matrix, filler, filler grade) and between the reaction kinetics, the network and the cross-linked structure was determined, which gives the opportunity to change the mechanical and the magnetic properties based on the analyzed impact factors and to expand the applications of bonded magnets in the field of thermoset-based ones.

**Keywords:** hard magnetic filler; highly filler thermosets; polymer-bonded magnets; viscosity





## 1. Introduction

The matrix material (thermoplastic, thermoset or elastomer) and a hard magnetic filler (e.g., strontium-ferrite-oxide (SrFeO) or neodymium-iron-boron (NdFeB)) are the main elements of a polymer-molded magnet [1]. Furthermore, additives are added in some cases to serve special demands in the application. For injection-molded magnets, a thermoplastic polymer is mainly chosen in terms of the matrix, and the filler content is limited by about 60% of the volume [2]. Due to the increasing viscosity, higher filler grades in the polymers can normally not be fabricated by injection molding [3]. Thermoset-based polymer-bonded magnets are primarily manufactured in a pressing process [4], where the filler can make up to 85% of the volume [2] and, with a higher percentage of filler within the matrix, higher magnetic properties can be realized within the pressing process [5]. The filler content defines the maximum magnetic properties a sample can have. As this content is higher in pressed samples, these parts have higher magnetic properties than injection-molded samples. The benefits of the injection-molding process are the freedom of design of the parts [5], the size accuracy [2], even for small and thin-shaped samples [6], and a functional integration, for example, by inserting metallic components [7]. The accuracy is realized due to less shrinkage in the material based on the high amount of fillers [8].

Polymer-bonded magnets are mainly applied in the fields of sensor and drive technology, whereby the main application within the drive technology is the magnetic excitation

of synchronous or direct current (DC) machines [9]. As yet, these applications are mainly realized by thermoplastic-based polymer-bonded magnets due to the fact that this type of magnet can be fabricated by injection molding and is therefore suitable for series production. However, thermosets enable a higher chemical permanence and thermal resistance as well as lower creep tendency with respect to thermoplastics. For example, the continuous operating temperature reaches 250 °C in terms of an epoxy resin but only 95 °C in terms of a polyamide. Furthermore, a thermoset molten mass reveals a significantly lower viscosity, which makes fillers more likely to move within the molten mass in the cavity [10]. Therefore, thermoset-based polymer-bonded magnets may have the opportunity to expand the applications for polymer-bonded magnets, for example, by realizing the demands of cooling water pumps or the chemical industry. A larger application field is especially given by the magnetic excitation of synchronous or DC machines, as chemical permanence and thermal resistance would allow the implementation in positions near the engine and pumps engaged with water or oil and within the chemical industry. It has to be taken into account that as yet, thermosets are mainly used in a pressing process. Therefore, the possibility of fabricating thermoset-based magnets in an injection-molding process has to be created to use the benefits of this fabrication method and to extend the application of polymer-bonded magnets.

### 1.1. Magnetic Properties

Magnetic properties are based on the smallest magnetic unit within a solid and can be described by the magnetic moment of a single electron and its rotation around its axis, also named spin [11]. With respect to the smallest magnetic unit, there are four different structural order systems with preferential direction by which the magnetic properties of materials are distinguished [12]. All magnetic moments are oriented parallel and reach the highest total magnetic moment for ferromagnetic materials. Those can be divided into hard and soft magnets. Hard magnets are also called permanent magnets as they have a characteristically high resistance against demagnetization [13].

The filler material differs in terms of the magnetic properties and geometries as well as particle sizes. For example, SrFeO particles can exhibit a hexagonal geometry with a particle size of 1–10 $\mu$m, and NdFeB commonly reveals a plate-like structure with a particle size of 100–400 $\mu$m. The resistance against demagnetization of SrFeO is two to three times lower than the resistance of NdFeB [14]. The filler material can result in isotropic or anisotropic magnetic properties [15]. The magnetic moments in isotropic filler particles are orientated randomly so that samples do not have a preferential direction regarding their magnetic properties. These properties are the best parallel to the preferential direction and the worst vertical to this direction in terms of anisotropic fillers, as the magnetic moments are orientated in a certain tack. Samples with anisotropic fillers reach for the remanence $B_r$, about 85% of the saturation flux density $B_s$ relative to the filler content and the quality of the production process. The remanence $B_r$ of the samples with isotropic fillers obtains only 50% of $B_s$ [5].

In terms of anisotropic fillers, the magnetic moments have to be oriented. The magnetization can be realized within the production or afterwards, for example, using impulse magnetization [3]. For anisotropic particles, the magnetic field strength influences the rate of the orientated particles and the possible remanence $B_r$ after magnetization until $B_s$. With that, a certain magnetic field strength is needed to guarantee a full orientation of anisotropic filler particles and the greatest possible magnetic properties with respect to the filler material [16]. A permanent magnet or an electromagnetic coil and a current through this conductor is used to orientate and magnetize the fillers in the process [3]. High magnetic properties can be reached by increasing the filler content or mixing different particle sizes so that smaller particles can fill the gaps between the greater ones. However, the orientation of the magnetic moments can be disturbed by particle interactions, which are more likely to occur if the filler content is high [17].

## *1.2. Injection Molding*

The behaviour of and the temperature setting for thermoplastics and thermosets in the injection molding process are completely different. Thermoplastics need a high temperature for plastification and are injected into the cavity with a low temperature leading to a rapid and premature cooling in the edge zone of the cavity along with a fast increase of the viscosity [18]. The temperature profile reveals a relatively high viscosity in the cavity, which impedes the orientation of the magnetic fillers [19]. Thermosets undergo a change of their chemical structure within the injection-molding process and under high temperatures, which are applied and lead to cross-linking of polymer chains. Therefore, plastification has to occur at a low temperature level to ensure that the temperature-driven process of curing takes place in the cavity. The high temperature in the cavity leads first to a reduction of the viscosity [20], which is assumed to be used to orientate filler materials evenly in the edge zone of the tool surface, and with that the magnetic field is displayed precisely. The time-dependent process of cross-linking increases the viscosity again with respect to the high temperature in the cavity [20].

## *1.3. Flow Behaviour and Reaction Kinetics of Thermosets*

The influence of filler material on the flow behaviour and reaction kinetics of thermosets has been examined in several papers [21–24]. It is therefore well known that the fillers affect the curing and rheological behaviour of epoxy resins, for example, by improving the thermal conductivity or modifying the chemistry of the cure [25]. The authors of [21] proved that below 100 °C the filler influences the reaction rate in terms of a silica filler with a mean diameter of 10 µm in an epoxy resin with a filler content of 30 vol.-%. Due to physical interactions between the resin and the filler, the reaction rate rises relative to the temperature. The free energy of the system is increased by the filler and enables the gelation to be more likely [26].

In [27], the influence of filler content, size and geometry on the thermal conductivity of a compound based on an epoxy resin is investigated. It was shown that the thermal conductivity increases with the filler content, especially with more than 50 vol.-%, due to the fact that the content of the particles is high enough to generate a mesh work of particles. Furthermore, a high amount of cuboid particles with a big particle improves the thermal conductivity within the processing of thermosets [27]. With respect to the results of [28], the flow behaviour can be enhanced by less filler content and increasing filler size. Furthermore, the cycle time can be reduced by a greater filler content and smaller filler size, taking into account that the filler size has less impact on both factors.

In [29], the gelation is defined as a function of the molecular network structure, which means that the influence of the fillers depends on the adhesive properties of the particles. If the particles portray no adhesive force to the resin, they act similar to holes in the network and reduce the gelation by interrupting the pathway. This is schematically shown in Figure 1. The trifunctional monomers of the resin join and build a tridimensional network during the curing. This network building can be interrupted by fillers, as shown in Figure 1A, with no adhesive force to the resin, leading to disturbed pathways of the network structure. If adhesive forces are present, the fillers enable the network structure to build around the fillers, integrating them in the curing process. This is portrayed in Figure 1B [29].

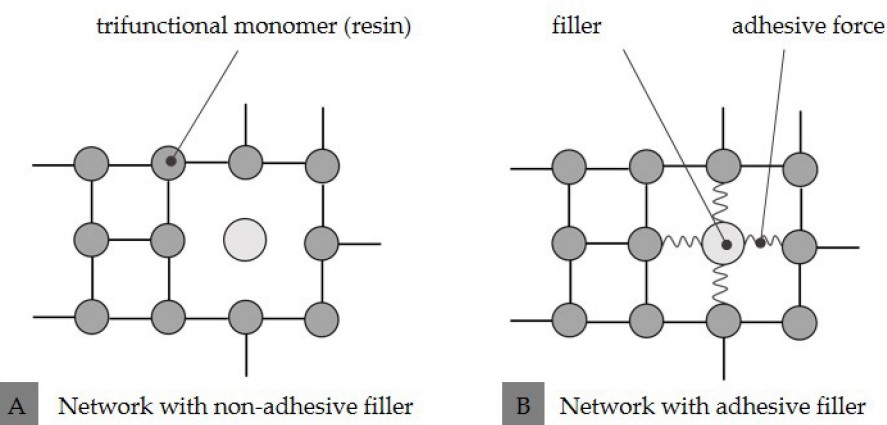

**Figure 1.** Network structure of filler and resin with no adhesive force (**A**) and adhesive force (**B**) between filler and resin [in parts: [29]].

In [30], the influence of hard magnetic particles on the flow and curing behaviour of highly filled thermoset-based compounds was investigated. Several factors were detected to affect the flow and curing behaviour of polymer-bonded magnets based on an epoxy resin (EP), whereby those factors can be applied to any filler system with a high filler amount in a thermoset compound due to the fact that they are based on the material behaviour of particles. Within this paper, it was shown that SrFeO acts like an adhesive filler in a network, being integrated, whereas NdFeB displays a network with a non-adhesive force, creating a hole in the network [30]. This theory of the filler behaviour in the network based on [29] is assumed to influence the viscosity of the compound systems, and with that a correlation to the magnetic and mechanical properties is likely.

However, most of this research so far is based on a relatively low filler content of less than 10 wt.-% or a highly filled system with respect to thermoset compounds with up to 50 wt.-%, whereby the results cannot be transferred automatically to highly filled systems in terms of polymer-bonded magnets with up to 90 wt.-%. The impact of hard magnetic fillers in thermosets has not been systematically studied. The authors of [31] showed that spherical particles with a particle size of 45 μm increase the flowability and curing velocity relative to plate-like fillers with a size of 100 μm. However, the influence of the structure and the size was not distinguished, and with that a precise insight of the influencing factors regarding hard magnetic fillers in a thermoset is not given.

The orientation of fillers with a preferential direction can only take place if the fillers are mobile and exhibit a magnetic moment [5]. So far, the correlation between the magnetic filler and its properties, the matrix material and the viscosity of the compound system on the magnetic properties has not been investigated. Furthermore, the impact of the fillers in the compound system and its mechanical properties have not been analyzed.

*1.4. Aim of the Paper*

This paper aims to investigate the influence of the matrix, the hard magnetic particles and the filler content on the viscosity of the compound and the impact on the magnetic and mechanical properties. In detail, the impact of two matrix materials (EP and phenolic resin (PF)) and two filler types (SrFeO and NdFeB) on the viscosity and the curing process is analyzed and correlated with the magnetic and mechanical properties. The basic understanding of the influence of the viscosity on these properties, as well as the impact of parameters on the viscosity, is essential for the fabrication of polymer-bonded magnets based on thermosets and the implementation of thermoset-based polymer-bonded magnets in new applications.

## 2. Materials and Methods

### 2.1. Material

The matrix material within the experiments was EP mixture of the type Epoxidur EP 368/1 (Raschig GmbH, Ludwigshafen, Germany) and a PF mixture of the type Resinol EPF 87120 (Raschig GmbH, Ludwigshafen, Germany). Both materials are a premixed black powder with resin, hardener, catalyst and some carbon black pigments. The exact composition of the mixture is a business secret of Raschig GmbH and therefore confidential. Table 1 reveals the important properties of the matrix material. The E-modulus and the tensile strength are based on the manufacturer's specifications, whereas the density and the heat capacity c are based on our own measurements.

**Table 1.** Specification of matrix material including density and heat capacity c (own measurements) as well as E-modulus and tensile strength (manufacturer's specifications).

| Matrix Material | Density (g/cm$^3$) | Heat Capacity c (J/(g·K)) | E-Modulus (GPa) | Tensile Strength (MPa) |
|---|---|---|---|---|
| Epoxy resin (EP) | 1.2250 | 1.616 | 13 ± 1 | 70 ± 10 |
| Phenolic resin (PF) | 1.2256 | 1.294 | 18 ± 2 | 130 ± 10 |

The experiments were conducted with the hard magnetic particles of anisotropic SrFeO and isotropic NdFeB. For SrFeO, the anisotropic type was OP71 (Dowa Holdings Co., Ltd., Tokyo, Japan), and for NdFeB the isotropic type was MQB+ (Magnetquench GmbH, Tübingen, Germany). The two filler types were investigated to compare two different network structures with (non) adhesive filler behaviour as well as the influence of the requirement of an external magnetic field for orientation in the case of the anisotropic filler. Table 2 presents the mean particle size (n: numerical; v: volumetric), density, thermal conductivity λ and heat capacity c based on our own measurements. The mean particle size of SrFeO is lower than that of NdFeB.

**Table 2.** Specification of filler material including mean particle size (n: numerical; v: volumetric), density, thermal conductivity λ and heat capacity c (own measurements) (SrFeO: strontium-ferrite-oxide; NdFeB: neodymium-iron-boron).

| Filler Material | Isotropy | Type | Mean Particle Size (µm) | | Density (g/cm$^3$) | Thermal Conductivity λ (W/(m·K)) | Heat Capacity c (J/(g·K)) |
|---|---|---|---|---|---|---|---|
| | | | n | v | | | |
| SrFeO | Anisotropic | OP71 | 4.87 | 1.94 | 5.38 | 2.3 | 0.639 |
| NdFeB | Isotropic | MQB + | 94.42 | 4.30 | 7.63 | 6.1 | 0.448 |

Figure 2 shows the filler-dependent geometry of the anisotropic filler material SrFeO (hexagonal) (Figure 2A) and isotropic NdFeB (plate-like) (Figure 2B) using a scanning electron microscope (Gemini Ultra-Plus; manufacturer: Carl Zeiss AG, Oberkochen, Germany).

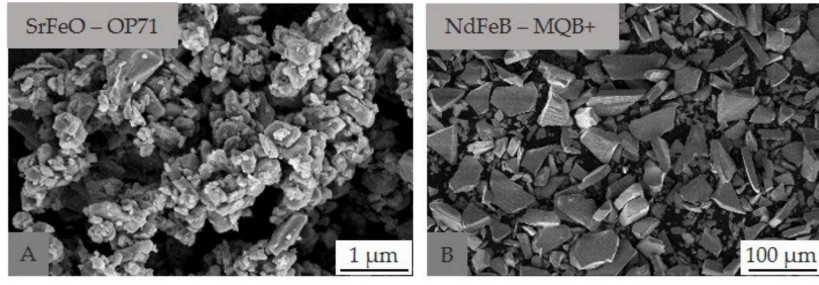

**Figure 2.** Particle geometry of anisotropic SrFeO (**A**) and isotropic NdFeB (**B**) using a scanning electron microscope (SrFeO: strontium-ferrite-oxide; NdFeB: neodymium-iron-boron).

For both filler types, the amount of the filler in the compound was kept constant at 50 vol.-% for the main experiments, which reveals a compromise between the magnetic properties and the flow conditions through highly filled compounds and particle–particle interactions. Furthermore, the filler content was varied between 50 vol.-% and 70 vol.-% in intervals of 5 vol.-% for the matrix material of EP and isotropic NdFeB as well as anisotropic SrFeO, to portray the influence of the filler content exemplarily.

### 2.2. Fabrication of the Test Specimens

To produce the compounds, the first preparation was conducted manually by mixing the two components in the dry state at room temperature. The proportion of the filler and the matrix material was weighted using a high-precision weighted device with respect to the amount of compound material needed for the fabrication of test specimens. After adding the exact amount of both components into a vessel, a bar was used to mix the compound manually. The monitoring of a homogeneous and sufficient mixing took only place by optical control.

Afterwards, the manual mixed compound was fabricated using a twin-screw extruder (Kraus Maffei Berstorff ZSE 25Ax45D, KrausMaffei Group, Munich, Germany) with a screw diameter of 25 mm and speed of 80 rotations per minute. The nozzle temperature was set at 90 °C to ensure that the material does not cure within the extruder. The test samples produced were pressure-controlled by a Krauss Maffei KM 80–380 CX DUR/03 injection-molding machine (KrausMaffei Group, Munich, Germany) with screw diameter of 30 mm. The processing parameters were set as shown in Table 3 without an outer magnetic flux density in the cavity. Within the dual cavity of the test samples of plates with the dimensions of 60 mm × 60 mm × 2 mm, a magnetic field was not integrated. By doing so, the magnetic properties were evaluated, which were realized by the injection molding process with respect to viscosity change by temperature and time but without the impact of an outer magnetic field. The mass temperature, the injection speed and the holding pressure were kept constant during the different material systems. The mold temperature was increased from 180 °C to 200 °C with rising filler content due to the different kinetics in the material system by adding particles. The heating time was changed in terms of the EP matrix, reaching 300 s instead of 120 s. This change shows an impact on the general reaction kinetics and is based on the different curing mechanism in EP and PF. The gel time is yet too long to reach economic standards, but optimization of the compound recipe was performed with respect to the viscosity.

**Table 3.** Processing parameters of injection molding to fabricate test samples (EP 245/1 with 40 to 70 vol.-% hard magnetic filler).

| | |
|---|---|
| Mass temperature | 85 °C |
| Mold temperature | 180–200 °C |
| Holding pressure | 250 bar |
| Heating time | 300 s (EP); 120 s (PF) |
| Injection speed | 15 mm/s |

### 2.3. Characterization

#### 2.3.1. Differential Scanning Calorimetry (DSC) According to DIN EN ISO 11357

To investigate the temperature-dependent reaction kinetics of the compounds, differential scanning calorimetry (DSC Q100, TA Instruments, New Castle, DE, USA) was used. Samples of about 5 mg were placed in DSC aluminum pans and heated with a constant rate of 10 °C per minute in terms of EP-based compounds, 5 °C per minute in terms of PF-based compounds and pure PF and 20 °C per minute in terms of pure EP. The different heating rates were chosen as the material systems revealed different sensitivities of the heating rate, mainly in terms of the reaction kinetic. The measurements were held from

0 °C to 300 °C. The experiments were conducted in a nitrogen atmosphere with a flow rate of 50 mL per minute. Figure 3 depicts the general route of a DSC measurement for the first heating cycle for thermoset materials as well as the two parameters, which were evaluated. The curing reaction is exothermal and can be characterized by the starting point ① with the corresponding temperature $T_1$ and time $t_1$ and the ending point ② with $T_2$ and $t_2$. The specific enthalpy $\Delta H_{ges;1}$ of the curing process as well as the peak temperature $T_{peak}$ were analyzed as they classified the curing process [32].

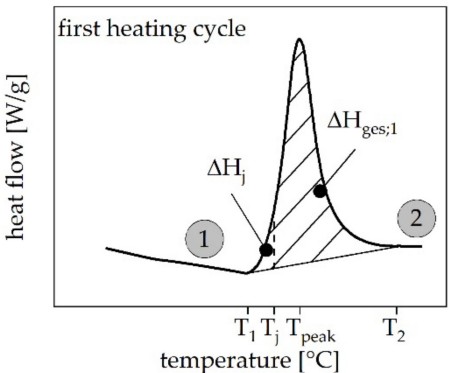

**Figure 3.** General route of a DSC measurement in the first heating cycle with characteristic parameters for the evaluation.

The reaction turnover $\alpha$ was calculated considering Equation (1), where $\Delta H_j$ is the specific enthalpy at the temperature $T_j$ and $\Delta H_{ges;1}$ is the total specific enthalpy in the first heating cycle [33].

$$\alpha = \left( \frac{\Delta H_j}{\Delta H_{ges;1}} \right) \cdot 100 \ [\%] \tag{1}$$

Furthermore, the cross-link density $\varphi_{link}$ was determined with respect to Equation (2), where $A_{un}$ is the uncured percentage of the material and with that the ratio between the total specific enthalpy in the second ($\Delta H_{ges;2}$) and the first ($\Delta H_{ges;1}$) heating cycle [33].

$$\varphi_{link} = 1 - A_{un} \ [\%] A_{un} = \left( \frac{\Delta H_{ges;2}}{\Delta H_{ges;1}} \right) \cdot 100 \ [\%] \tag{2}$$

### 2.3.2. Determination of the Viscosity Using a Rotational Viscometer According to DIN EN 6043

To characterize the viscosity with respect to an increasing temperature (dynamic behaviour) and to the time dependence (isothermal), a rotational viscometer (Discovery Hybrid Rheometer 2, TA Instruments, New Castle, DE, USA) was used according to DIN EN 6043. The assembly was based on two plates with a shearing load. The frequency was held constant at 1 Hz. In the case of the dynamic measurements, a start temperature of 70 °C for NdFeB filler and 100 °C for SrFeO was set up. The granule was placed between two plates and the distance between them was reduced to the starting position. After reaching the starting setup, the shell was placed around the stamps and floated with nitrogen. The measurement was carried out with a constant heating rate of 5 K/min and stopped at 200 °C after the viscosity had not changed further. The minimum of the viscosity $\eta_{min}$ was analyzed.

The isothermal measurements started at a certain temperature, which was held constant, and the change of the viscosity depending on the time was determined. The isothermal plateau was set at the temperature of $\eta_{min}$, which was 110 °C in the case of PF and 120 °C in the case of EP, and increased in steps of 10 °C up to 140 °C. Furthermore, the temperature of 160 °C was analyzed. The route of the viscosity $\eta$ relative to the time follows

an s slope. The time $t_{tp}$ between the beginning of the calculation and the turning point was analyzed.

### 2.3.3. Mechanical Properties according to DIN EN ISO 527

The mechanical properties were analyzed using bar-shaped samples, which were prepared out of the plates with the dimensions of $60 \times 10 \times 2$ (mm$^3$) with a milling machine. As the compound reveals a brittle behaviour, the preparation of tensile bars was not possible. The properties were determined using a universal tensile testing machine (type: 1464, ZwickRoell GmbH & Co. KG, Ulm, Germany) with a traverse speed of 0.3 mm/min. The analysis was held at standard climate of 23 °C and 50% relative humidity. Beside the stiffness or representative, the E-modulus $E_t$, the tensile strength $\sigma_m$ and the elongation at break $\varepsilon_m$ were defined.

### 2.3.4. Magnetic Properties

The samples for the determination of the magnetic properties were prepared from the plates using a saw. The test samples with the dimensions of $20 \times 20 \times 2$ (mm$^3$) were extracted from the middle of the plate. The magnetic properties represented by the remanence $B_r$ were evaluated by measuring the hysteresis loop using a permagraph (type: C-300, Magnet-Physik Dr. Steingroever GmbH, Cologne, Germany). A magnetic field strength H from outside is applied onto the samples to magnetize and demagnetize the material within the permagraph. A sensor gathers the magnetic flux density B or the magnetic polarization relative to magnetic field strength H. The magnetic field was related perpendicular to the quadratic interface of the sample. Before each measurement, a pulse magnetizer (type: Im-12220-U-MA-C, Magnet-Physik Dr. Steingroever GmbH, Cologne, Germany) and a magnetic device (type: MV D30·30 mm F-TC, Magnet-Physik Dr. Steingroever GmbH, Cologne, Germany) were used to ensure a full magnetization of the filler particles. The capacitor bank of the pulse magnetizer was charged by a transformer till 1.9 kV. This is sufficient to fully magnetize the filler particles. The stored energy was discharged by a thyristor within the coil of the magnetic device. All measurements were conducted at room temperature.

### 2.3.5. Filler Orientation

To analyze the filler orientation, the samples were embedded in cold-curing epoxy resin (type: Epofix, Struers GmbH, Ottensoos, Germany). The specimens were then split in the centre using a water-cooled saw with minimal temperature input so that microscopic examinations could be performed in the centre of the specimen and perpendicular to the expected long axis of the filler. The specimens were further demagnetized and polished.

In the case of NdFeB, the orientation was characterized by a stereo microscope (type: Axio Zoom.V16, Carl Zeiss AG, Oberkochen, Germany). As the mean particle size of SrFeO is significantly lower relative to NdFeB, a scanning electron microscope (type: Gemini Ultra-Plus, Carl Zeiss AG, Oberkochen, Germany) was used to determine the orientation. Here, a 10 nm layer of spray gold was placed on top of the samples. On the basis of the images taken by the stereo or the scanning electron microscope, a differentiation between the matrix material and the filler was carried out by means of a grey scale threshold analysis. In order to determine the main orientation angle between 0° and 90°, the orientation was evaluated along the longest axis of the individual particles. According to the formation of the histogram, a reduced orientation due to a broad scattering of the histogram or a preferred orientation of the fillers could be deduced. The orientation between 0° and 15° is the expected one due to the flow condition [34] and further improves the magnetic properties, taking the preferred direction in anisotropic particles into account. The orientation between 15° and 75° is set as stray field and should be minimized. Between 75° and 90°, the oriented fillers have no impact on the magnetic field and should therefore be oriented again, as low as possible. The distribution of the filler orientation within 0° and 90° is shown in Figure 4. The filler orientation in this process could only be influenced by the shear and flow

conditions based on the viscosity of the compound as well as possible particle interactions due to the missing outer magnetic field.

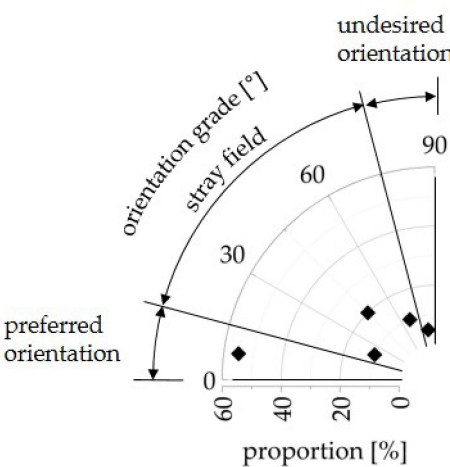

**Figure 4.** Ideal orientation grade distribution with high proportion in terms of the preferred orientation and low proportion in the other two sectors.

## 3. Results and Discussion

### 3.1. Differential Scanning Calorimetry (DSC)

The route of the DSC measurements (A) and the specific enthalpy $\Delta H_{ges;1}$, as well as the peak temperature $T_{peak}$ (B) with respect to different material systems, are shown in Figure 5 for the constant filler grade of 50 vol.-% in comparison to the pure matrix material. The specific enthalpy needed for the curing process is significantly reduced with fillers compared to the pure matrix material and slightly lowered in terms of the PF matrix material. Furthermore, $T_{peak}$ is hardly changed by the filler, but the level increases for the EP matrix relative to PF.

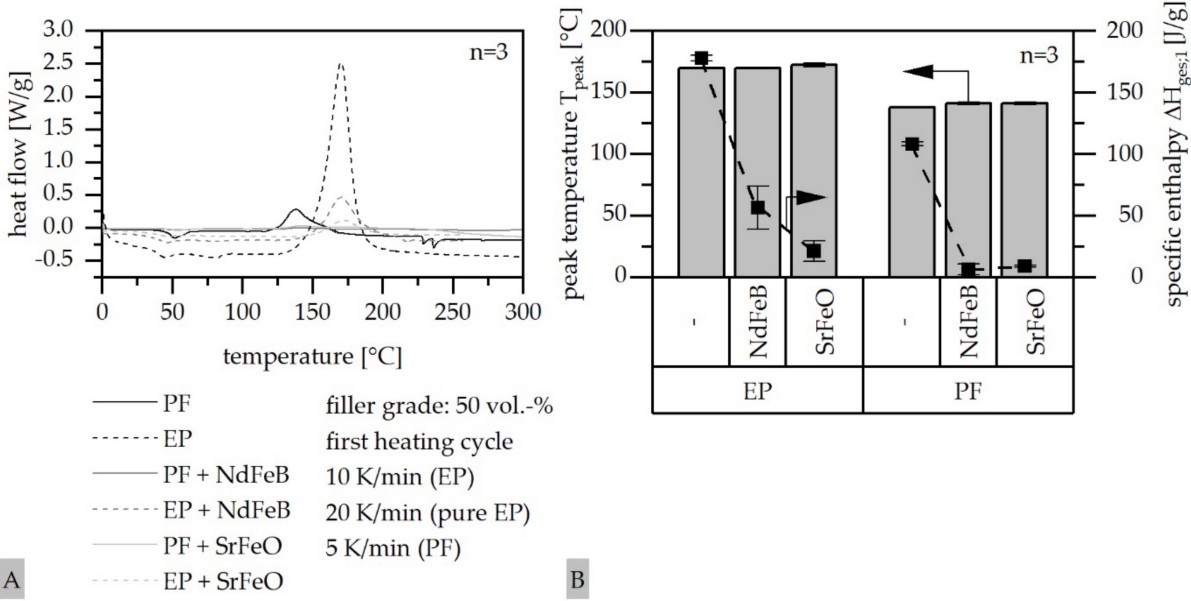

**Figure 5.** Route of DSC measurement (**A**) and peak temperature $T_{peak}$ as well as specific enthalpy $\Delta H_{ges;1}$ (**B**) in the first heating cycle relative to the material system (SrFeO: strontium-ferrite-oxide; NdFeB: neodymium-iron-boron; PF: phenolic resin; EP: epoxy resin).

As the filler itself is not reactive in the curing process, the amount of material that can be involved in this process is reduced with increasing filler amount. Taking into account

the exothermal heat flow, the specific enthalpy is reduced due to the presence of filler material. The levels of the specific enthalpy $\Delta H_{ges;1}$ and the peak temperature $T_{peak}$ are mainly influenced by the matrix material in terms of the different heat capacity c. With that, PF requires less applied heat to reach higher temperatures due to a low heat capacity c compared to EP. This leads to lower $T_{peak}$ in terms of PF with respect to a constant heat rate. The filler itself has only a small impact on $\Delta H_{ges;1}$ due to the different thermal conductivity $\lambda$ of the two types of filler. Furthermore, the filler amount reduces $\Delta H_{ges;1}$ without changing $T_{peak}$. With that, the thermal conditions for the curing process are mainly defined by the chosen matrix material and slightly by the filler type.

To analyze the curing process, the reaction turnover $\alpha$ relative to the temperature is shown in Figure 6 for different matrix materials and fillers (A) with a constant filler grade of 50 vol.-% and for different filler grades in terms of EP with NdFeB (B). The S-shaped route of $\alpha$ is shifted to higher temperatures in terms of fillers with a greater impact by SrFeO and the EP matrix material. This means that, for example, 60% of the reaction turnover $\alpha$ is reached at about 140 °C in terms of pure PF but only at 150 °C in terms of PF with 50 vol.-% NdFeB and 175 °C in terms of EP with 50 vol.-% NdFeB. This goes along with the reduced peak temperature $T_{peak}$ of PF-based compounds. The impact of the filler grade on the reaction turnover $\alpha$ is low, as the temperature at which a certain $\alpha$ is reached is only slightly changing throughout the increase of the filler grade between 50 vol.-% and 70 vol.-%.

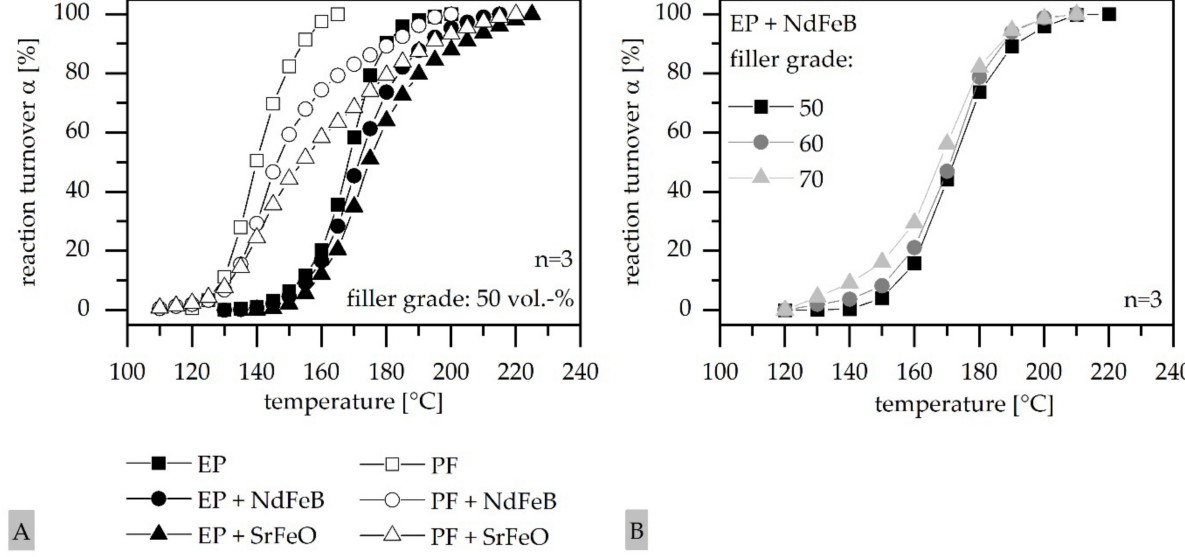

**Figure 6.** Reaction turnover $\alpha$ relative to the temperature for different matrix materials and fillers with constant filler grade of 50 vol.-% (**A**) and for different filler grades in EP with NdFeB (**B**) (SrFeO: strontium-ferrite-oxide; NdFeB: neodymium-iron-boron; PF: phenolic resin; EP: epoxy resin).

The matrix material and the filler type influence the reaction turnover $\alpha$ highly, whereas the filler grade has only a small impact. The shift of the route further goes along with a higher time dependence through highly filled compounds, the EP matrix and SrFeO. This is based mainly on the thermal conductivity $\lambda$ and the heat capacity c of the fillers. In terms of NdFeB, the thermal conductivity $\lambda$ is higher and the heat capacity c is lower compared to SrFeO. Due to the different heat capacity c in the matrix and filler as well as the high $\lambda$ in terms of NdFeB, more energy is needed for the temperature change and the cross-linking process. The applied heat is less concentrated in the filler areas due to the high $\lambda$. As the thermal conductivity $\lambda$ is high in terms of NdFeB, faster curing takes place. Furthermore, the slope of the two pure matrix materials is similar, but the gradient changes significantly for more than 50% of $\alpha$ in terms of the compound. The slope of PF in the compound is reduced compared to EP compounds, which is assumed to occur through

less coupling of the filler in terms of the PF matrix. The influencing factors on the three parameters in terms of the reaction kinetic $\Delta H_{ges;1}$, $T_{peak}$ and $\alpha$, as well as the causes, are summarized in Table 4.

**Table 4.** Influencing factors and causes on the specific enthalpy $\Delta H_{ges;1}$, the peak temperature $T_{peak}$ and the reaction turnover $\alpha$ (++: high impact; +: low impact; 0: no impact).

| Parameter | Influencing Factor | | | Cause |
|:---:|:---:|:---:|:---:|:---:|
| | matrix | filler | filler grade | |
| Specific enthalpy $\Delta H_{ges;1}$ | ++ | + | + | Heat capacity c of the matrix; thermal conductivity $\lambda$ of the filler |
| Seak temperature $T_{peak}$ | ++ | 0 | 0 | Heat capacity c of the matrix |
| Reaction turnover $\alpha$ | ++ | ++ | + | Heat capacity c and coupling of the matrix; thermal conductivity $\lambda$ of the filler |

### 3.2. Change of the Viscosity Relative to Matrix, Filler and Filler Amount

The influence of the matrix and filler as well as the filler grade on the viscosity is shown in terms of the dynamic measurement in Figure 7A and in dependence to the time at different but constant temperature levels in Figure 7B. The minimum of viscosity $\eta_{min}$ increases exponentially with the filler grade, but SrFeO reveals a greater impact, and with that the slope is steeper compared to NdFeB. In general, the compound based on EP reveals lower $\eta_{min}$ compared to PF, as well as NdFeB compared to SrFeO. The isothermal measurements define a higher time at the turning point $t_{tp}$ in the case of the EP and NdFeB (independent to the matrix system), but the matrix system reveals a higher impact. The route of $t_{tp}$ is exponentially decreased with increasing temperature, and the input of the matrix or filler decreases with higher temperatures.

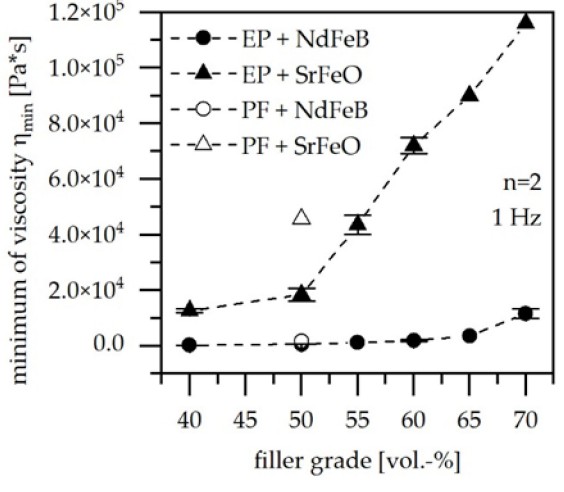

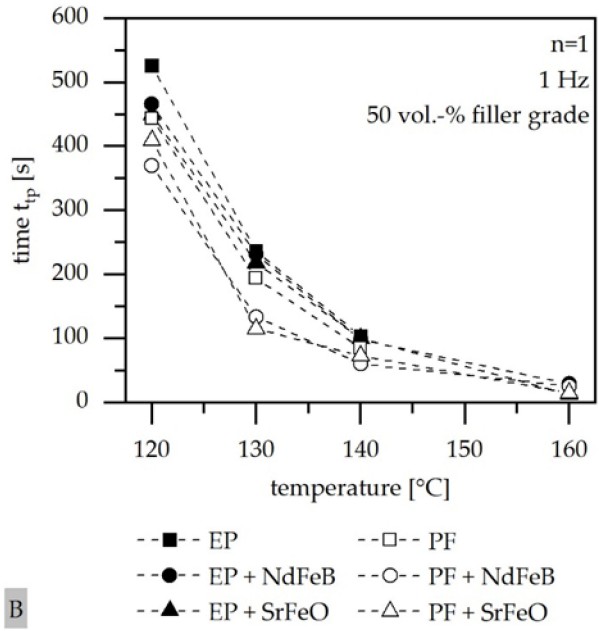

A

B

**Figure 7.** Minimum of viscosity $\eta_{min}$ (**A**) and time at turning point $t_{tp}$ relative to the temperature level (**B**) for different matrix materials, fillers and filler grades (SrFeO: strontium-ferrite-oxide; NdFeB: neodymium-iron-boron; PF: phenolic resin; EP: epoxy resin).

It is assumed that the network structure causes the difference in the minimum of the viscosity $\eta_{min}$, as SrFeO is more likely to be integrated into the network, and with that it impedes the flow behaviour leading to higher $\eta_{min}$ in terms of material systems with SrFeO. Furthermore, pure PF shows a poor cross-link structure (compare Section 3.3 with high

uncured percentage $A_{un}$ in the case of pure PF) so that a low number of the crosslinking points already increases the viscosity significantly. This leads to higher values in terms of $\eta_{min}$ for PF-based compounds and shorter $t_{tp}$. With that, PF-based compounds are more time-dependent and reveal less time before curing. The influencing factors on the two parameters of the viscosity $\eta_{min}$ and $t_{tp}$, as well as the causes, are summarized in Table 5.

**Table 5.** Influencing factors and causes on the minimum of viscosity $\eta_{min}$ and the time of the turning point $t_{tp}$ (++: high impact; +: low impact; 0: no impact).

| Parameter | Influencing Factor | | | Cause |
|:---:|:---:|:---:|:---:|:---:|
| | matrix | filler | filler grade | |
| Minimum of viscosity $\eta_{min}$ | + | ++ | ++ | Network structure |
| Time at the turning point $t_{tp}$ | + | + | —— | Cross-link density $\varphi_{link}$ and structure |

The network structure and the cross-link structure are essential parameters in terms of the viscosity and with that the properties of samples. Therefore, Figure 8 depicts the assumed structure for both the matrix and filler systems analyzed in this paper. The classification is based on the general theory of [29] and mainly, in terms of the thermoset-based compounds, on our own measurements, further described in [30] in terms of the filler. As illustrated, EP reaches a high cross-link density $\varphi_{link}$ compared to PF, and SrFeO is (partly) integrated in the matrix, whereas NdFeB is loose in the network structure without adhesive forces.

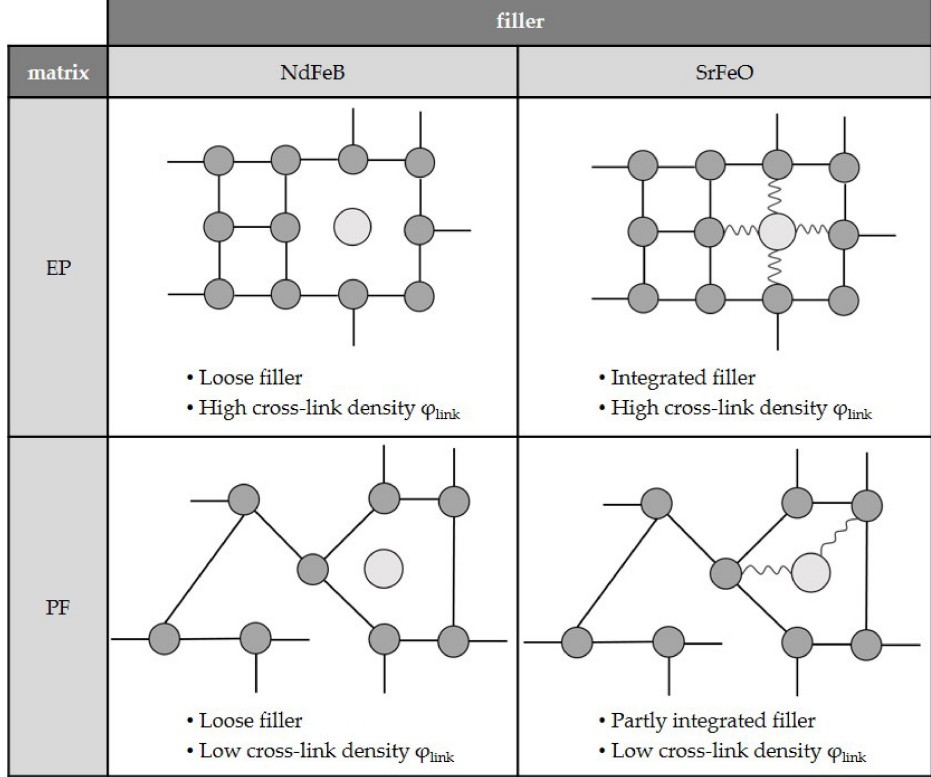

**Figure 8.** Network and cross-link structures in terms of the matrix material and the filler (SrFeO: strontium-ferrite-oxide; NdFeB: neodymium-iron-boron; PF: phenolic resin; EP: epoxy resin).

### 3.3. Influence of the Reaction Kinetics and the Viscosity on the Mechanical Properties Relative to the Material System

The mechanical properties mainly depend on the network structure and the cross-link density but further go along with the general material properties such as the heat capacity c and the thermal conductivity λ. In general, the E-modulus $E_t$ and the elongation at break $\varepsilon_t$ are correlated with the network structure [35], whereas the tensile strength $\sigma_m$ depends on the cross-link density [36]. In case of a low value of the uncured percentage $A_{un}$, the cross-link density is high, and the material behaviour is brittle with a small value of the tensile strength $\sigma_m$. According to Figure 8, a low $\sigma_m$ can be expected in the case of PF-based materials. Figure 9A depicts the mechanical properties, namely the E-modulus $E_t$, the tensile strength $\sigma_m$ and the elongation at break $\varepsilon_m$ with respect to different matrix and filler materials. Furthermore, the uncured percentage is shown in Figure 9B in the case of the different compounds and the pure matrix material. The SrFeO-based compounds reveal a high $E_t$ and a low $\varepsilon_m$ compared to NdFeB-based compounds. NdFeB-based compounds depict a low $A_{un}$ as well as low $\sigma_m$, which goes along with the assumed behaviour. However, SrFeO-based compounds show a contrary behaviour between $A_{un}$ and $\sigma_m$. Furthermore, the reached level of $A_{un}$ in the pure matrix and the mechanical properties regarding the manufacturer specifications fits well with respect to Table 1.

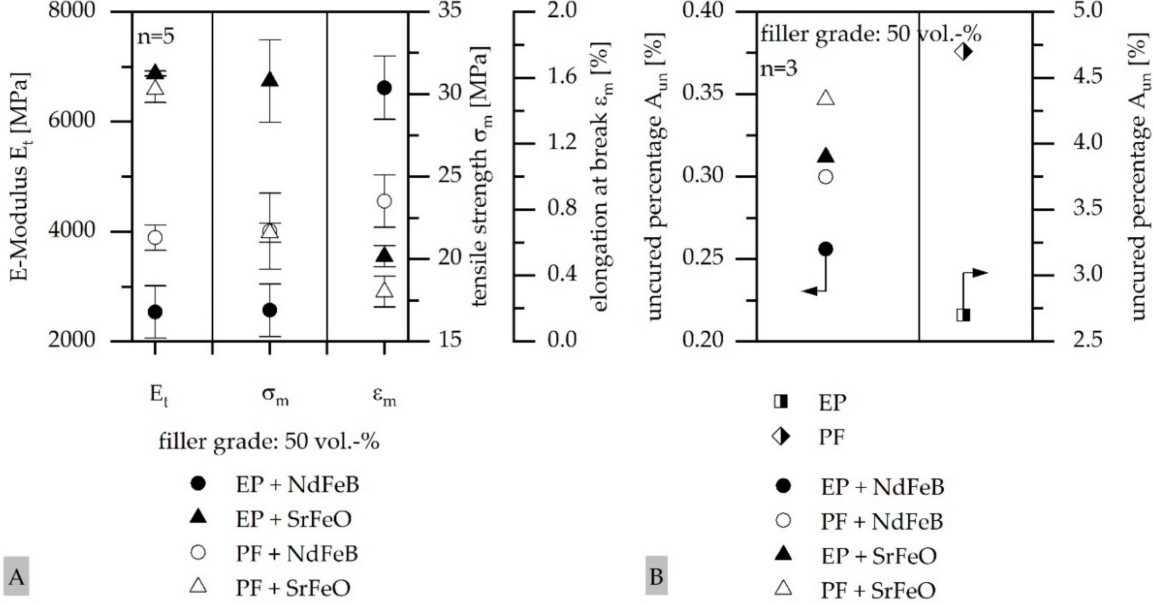

**Figure 9.** Mechanical properties (**A**) and uncured percentage $A_{un}$ (**B**) relative to different matrix materials and fillers (SrFeO: strontium-ferrite-oxide; NdFeB: neodymium-iron-boron; PF: phenolic resin; EP: epoxy resin).

As SrFeO is integrated in the network structure due to adhesive forces, the elastic proportion is low, which leads to higher values of stiffness and less elongation after break. The contrary behaviour can be seen in the case of the NdFeB filler with no integration. In the case of SrFeO, in terms of the tensile strength $\sigma_m$, the filler and in particular the filler's size acts as a crack initiation, leading to low $\sigma_m$ in terms of the PF matrix. In the EP matrix, SrFeO is well integrated, which leads to a fine-grained microstructure and a stiff behaviour with higher $\sigma_m$.

With an increasing filler content $E_t$, $\sigma_m$ increases in terms of SrFeO in an EP matrix with respect to the realized microstructure. In NdFeB-based compounds with an EP matrix, $E_t$ is first reduced but increases again above 60 vol.-%. This is assumed due to higher packing density, which inhibits the mobility of the fillers in the network. Furthermore, $\sigma_m$ is reduced slightly with increasing filler content. The impact of the filler grade on the

mechanical properties, namely E-modulus $E_t$ and tensile strength $\sigma_m$, is shown in Figure 10. As the elongation after break $\varepsilon_m$ is changed, similarly to $E_t$ but inverted at the ordinate, the route is not further portrayed.

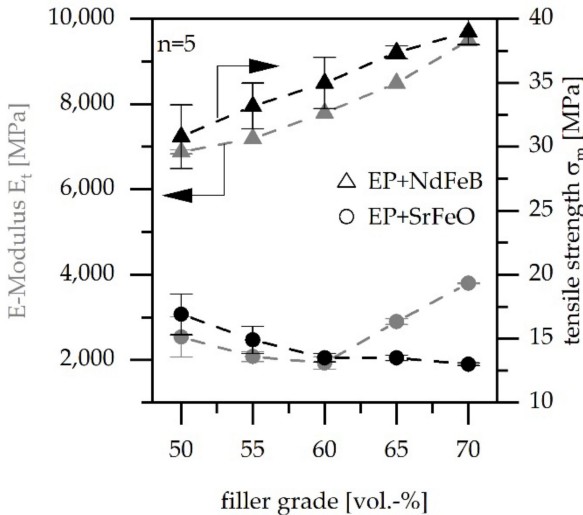

**Figure 10.** Mechanical properties of EP-based compounds relative to different fillers and filler grades (SrFeO: strontium-ferrite-oxide; NdFeB: neodymium-iron-boron; EP: epoxy resin).

*3.4. Influence of the Reaction Kinetics and the Viscosity on the Magnetic Properties Relative to the Material System*

In addition to the mechanical properties, the magnetic properties rely on the network and cross-link structure as well. The magnetic properties in terms of different matrix and filler systems are shown in Figure 11A, as well as in terms of the filler grade in Figure 11B, which is exemplary for NdFeB in an EP matrix. The remanence $B_r$ increases in terms of the PF matrix relative to the EP matrix independent to the filler system. Furthermore, NdFeB reaches higher values for $B_r$ due to the fundamental magnetic properties of the filler system. Due to higher filler grades, the magnetic properties increase significantly up to 60 vol.-% but stagnate with filler grades higher than 60 vol.-% due to an inhibition of the orientation because of the high filler amount. This amount makes particle interactions more likely.

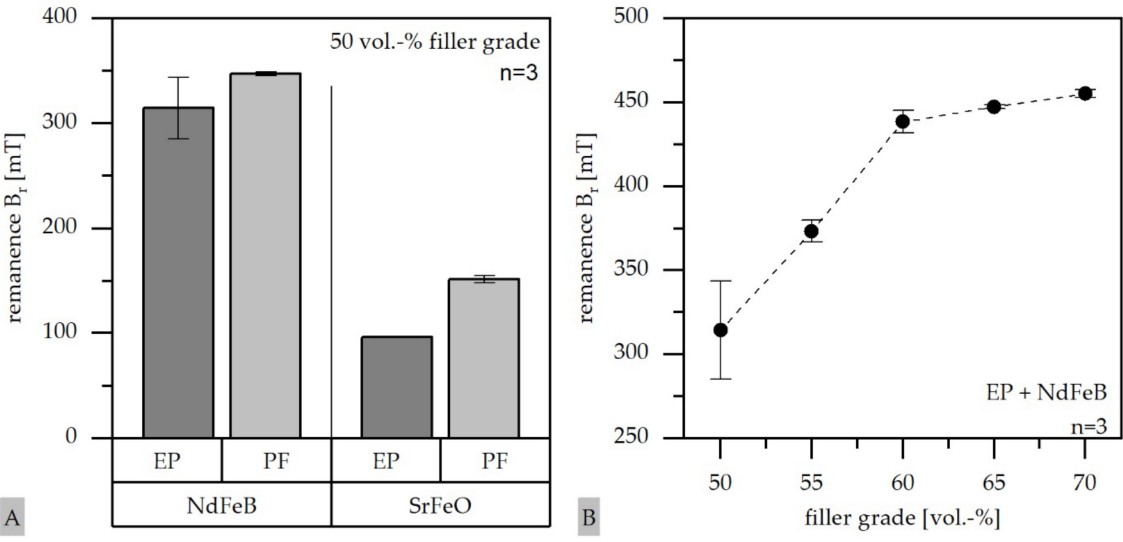

**Figure 11.** Magnetic properties relative to different matrix materials and fillers (**A**) and filler grades in EP with NdFeB (**B**) (SrFeO: strontium-ferrite-oxide; NdFeB: neodymium-iron-boron; PF: phenolic resin; EP: epoxy resin).

Figure 12 shows the orientation relative to the different matrix and filler systems with a constant filler grade of 50 vol.-% (Figure 12A) and a varying filler grade between 50 and 70 vol.-% representatives in EP with NdFeB (Figure 12B). The orientation in the preferred direction (between 0° and 15°) is increased in terms of the NdFeB relative to SrFeO. Furthermore, the difference between the two matrix systems is similar in terms of NdFeB but is increased for PF in terms of SrFeO. The proportion of the orientation between 75° and 90° is similar in all material systems. In the stray field (between 15° and 75°), SrFeO reveals a higher amount, especially in terms of EP. The impact of the filler grade is low between 50 and 60 vol.-% but increases in terms of 70 vol.-%. Here, the proportion of the preferred orientation is reduced and that of the stray field is increased.

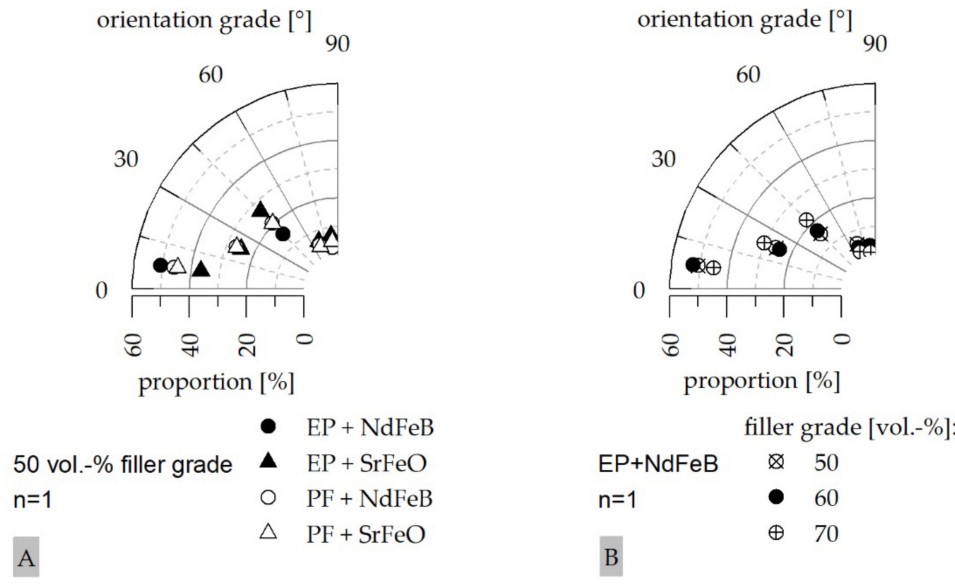

**Figure 12.** Orientation grade relative to different matrix materials and fillers (**A**) and filler grades in EP with NdFeB (**B**) (SrFeO: strontium-ferrite-oxide; NdFeB: neodymium-iron-boron; PF: phenolic resin; EP: epoxy resin).

The low cross-link structure in the PF matrix allows the filler particle to move freely, which leads to higher magnetic properties even without an outer magnetic field. With that, the structure in PF allows for fillers more likely to orientate due to the shear forces, which can further be used by an outer magnetic field application. This effect can especially be seen in terms of SrFeO, which reveals a high cross-link structure in terms of EP. Therefore, not only is the proportion in the preferred direction reduced, but the proportion in the stray field is increased further. It can be predicted that higher magnetic forces are needed to overcome this hindrance in order to orientate anisotropic fillers. The reduced increase of the magnetic properties with filler grades higher than 60 vol.-% is assumed to be caused by particle interactions, which hinder a homogenous orientation and with that an increase in the magnetic properties. This can also be seen with respect to the filler orientation analyses.

## 4. Conclusions

It was shown that the reaction kinetics of the hard magnetic and thermoset-based compounds are influenced by the matrix material, filler type and grade with respect to the heat capacity c and the thermal conductivity λ. The viscosity is based on these material properties as well as on the network and cross-link structure. The two investigated matrix materials (EP and PF), as well as the fillers (SrFeO and NdFeB), could be related to the network structure theory, which is based on [29], but extended with respect to the determinations portrayed in this paper and our own measurements. Based on this theory, the mechanical and magnetic properties were discussed and correlated to the reaction kinetics and viscosity behaviour. The general change in the behaviour with respect to

the reaction kinetics, the viscosity and the mechanical and magnetic properties relative to the impact through the matrix material, the filler and the filler grade is summarized in Figure 13. Note that, for example, the material system causes contrary shift directions in the reaction kinetics and the viscosity. Furthermore, a low viscosity itself does not ensure high magnetic properties, as the network structure further influences the mobility of the fillers within the matrix system.

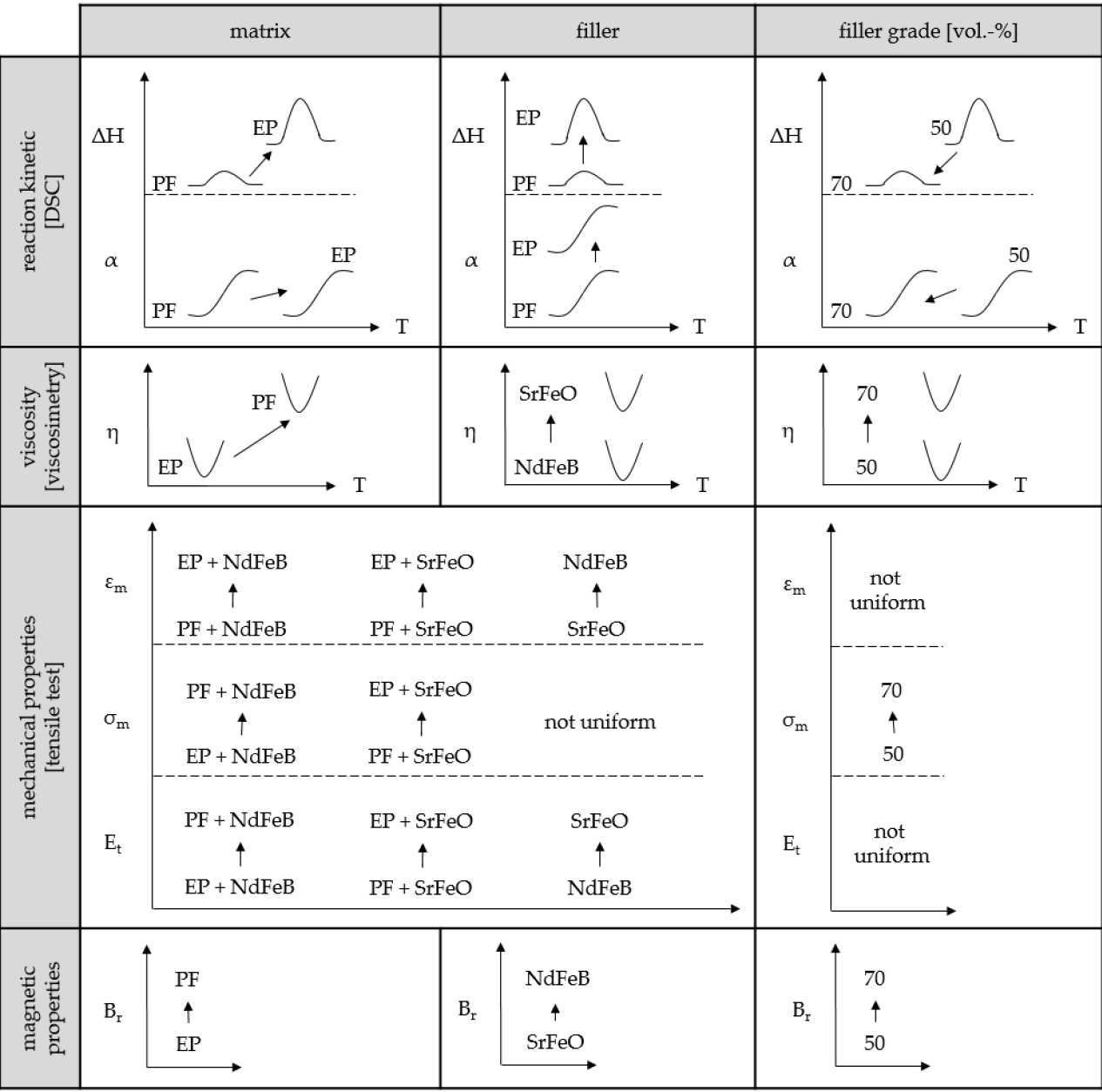

**Figure 13.** Overview of the impact of matrix, filler and filler grade on the reaction kinetics, the viscosity, the mechanical and magnetic properties and the correlation between the parameters (SrFeO: strontium-ferrite-oxide; NdFeB: neodymium-iron-boron; PF: phenolic resin; EP: epoxy resin).

The presented investigations show the basic connection between the material and the reaction kinetics as well as viscosity. Furthermore, the relation to the mechanical and magnetic properties is explained. An evaluation of the achieved properties, also with regard to possible applications, is examined and evaluated in following research work. Further investigations will be carried out in terms of the impact of an outer magnetic field

on the orientation of fillers and the separation of the effects in terms of the orientation by shear and flow conditions and by an outer magnetic field. This will further be correlated with the reaction kinetics and the viscosity as well as the network and cross-link structure.

**Author Contributions:** U.R.: conceptualization, methodology, validation, investigation, writing—original draft and visualization; D.D.: writing—review and editing, supervision and project. All authors have read and agreed to the published version of the manuscript.

**Funding:** This research was funded by the German Research Foundation (DFG) within the project DFG DR 412/36-1 "Duroplastgebundene spritzgegossene Dauermagneten mit definierter Magnetisierungsstruktur". We acknowledge financial support by Deutsche Forschungsgemeinschaft and Friedrich-Alexander-Universität Erlangen-Nürnberg within the funding programme "Open Access Publication Funding".

**Data Availability Statement:** Restrictions apply to the availability of these data. Data are available with the permission of the author.

**Acknowledgments:** We want to thank the German Research Foundation (DFG) for funding the project DFG DR 412/36-1 "Duroplastgebundene spritzgegossene Dauermagneten mit definierter Magnetisierungsstruktur" as well as Friedrich-Alexander-Universität Erlangen-Nürnberg for the funding programme "Open Access Publication Funding".

**Conflicts of Interest:** The authors declare no conflict of interest. The funders had no role in the design of the study; in the collection, analyses or interpretation of data; in the writing of the manuscript or in the decision to publish the results.

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
