# Peer review of "Understanding the Effect of Material Parameters on the Processability of Injection-Molded Thermoset-Based Bonded Magnets"

_2673-8724, doi:10.3390/magnetism2030016_

Round 1

Reviewer 1 Report

General comments about the manuscript:

The manuscript entitled “Understanding the effect of material parameters onto processability of injection molded thermoset based bonded magnets” describes the preparation of thermoset based polymer bonded magnets based on strontium-ferrite-oxide SrFeO or neodymium-iron-boron NdFeB. In my opinion, the study is very complete and the work well motivated. In general terms, the manuscript is very well-written, and including very illustrative figures, schemes and summary tables. Therefore, I would recommend it acceptance with minor changes.

Specific changes suggested:

First, I would like to apologize in advance because my knowledge in polymers is rather limited and therefore, some of my comments might sound rather basic to the authors and they probably have trivial answers. Nonetheless, I believe many of the potential readers of this work may come from the purely magnetic world as it is my case, so I would still encourage the authors to include some info in this regard to make the article accessible for a broader audience. Besides this, I also have some small comments and suggestions on the manuscript.

All my suggestions/comments (related and unrelated to polymers) are listed below:

- Page 1: I would recommend mentioning in the abstract the fillers studied in the work (SrFeO, NdFeB)

- Lines 12-14: Consider rephrasing. Unfortunately, I cannot make full sense out of the sentence: “Based on the determination of the impact, the theory of the network structure is found as the network and the cross-linked structure define not only the material but further the sample behaviour significantly.”

- Lines 47-48: authors explain how thermosets yield higher chemical permanence and thermal resistance. It would be useful to provide approximate numbers on the temperatures they can stand.

- Line 70-71: SrFeO particles are often platelets too – there are many examples of this in the literature (eg https://doi.org/10.1016/j.matdes.2017.03.082) – and both phases can have sizes below the states in the text. Try to rephrase this sentence to make it a bit more general.

- Lines 94-107: Can you be more specific with which temperature values (ranges) you refer to when you speak about high/low temperatures?

- Lines 99-100: You write “Thermosets undergo a change of the chemical structure within the injection molding process” is it a chemical change (i.e. phase change, secondary phases) or a structural change (i.e. strain induced in the structure, phase transition but same chemical composition)?

- Line 184: there is something strange with this table, at least in the pdf file I received. Some values seem duplicated and overlapping with others.

- Lines 226-237: what type of pressure is induced by an injection molding machine? Is it uniaxial or isotropic? Can you give some additional details on this?

- Lines 235-6: what is the reason behind increasing the mold temperature with raising the filler content?

- Figure 4: What do the diamonds correspond to? Are they samples prepared in this work or just examples? In my opinion, they are not really necessary at this point when authors are explaining how to interpret these orientation diagrams. In any case, authors may update the caption accordingly.

- Lines 413-437 and Figure 8: Authors do not seem to justify why SrFeO is more likely to be integrated in the network than NdFeB or the reasoning behind the assumed network structures depicted in Figure 8. They only state “The classification is based on the general theory of [29]”. However, this classification seems quite relevant, as later on, the mechanical properties are understood based on this. As I mentioned earlier, I am not in this field and therefor I apologize in advance if this is rather evident for those familiar with polymers and injection molding, but still, it might not be evident for the magnetic community – as it isn’t for myself. Therefore, I believe the story would benefit from having a bit information on this matter.

- Lines 457-458: there seems to be a spare line break here

-- Page 14: I believe this table at the bottom is a repetition of Table 1.

- Line 491: Br does not increase further for filler grades > 60 vol%. I presume this is due to a worse orientation and will be explained later in the text but I would suggest briefly bringing it up already at this point, otherwise the trend feels a bit “unexplained”.

- Figure 12: If possible, I would recommend plotting the values with error bars, it would help judging whether small differences between samples are meaningful of just within the uncertainty of the calculation.

- Although the summary table on Figure 13 gives a very nice overview of the study, I do miss a broader discussion that compares the results obtained in this study with the previous literature, in terms of both magnetic and mechanical properties. How do your values compare?

- Additionally, I have detected a few typos along the document:

Line 46: “mmagnetcan”, Line 48: “inwithespect”, Line 233: “,and”, Line 297: replace “carry” by “carried”,  Figure 13 top left: “kincetic”.

Author Response

Dear Sir or Madam,

thanks a lot for your time to review my article. I really appreciate your feedback regarding my work. Please find attached my response to your review. 

Kind regards 

Uta Rösel 

Reviewer 2 Report

The manuscript "Understanding the effect of material parameters onto processability of injection molded thermoset based bonded magnets"  by  Uta Rösel and  Dietmar Drummer shows how the reaction kinetics of the hard magnetic and thermoset based 

compounds are influenced by the matrix material, filler type and grade with respect to the heat capacity and the thermal conductivity. The two matrix materials (EP and PF) as well as the fillers (SrFeO and NdFeB) are considered.

It should be noted, that the fabrication of thermoset based magnets in an injection molding process is important task which could extend the application of polymer bonded magnets. 

I could recommend the article for publication after the following details will be clarified. 

The methods of structurization and improvement of SrFeO and NdFeB magnets mentioned in the Introduction by providing the citation of the Ref[1-14], it will be more instructive if magnetic structures operating with this permanent magnets for creation of bias magnetic field was also mentioned in this list and citation of Refs[Bulletin of the Russian Academy of Sciences: Physics 85 (6), 595-598 (2021)], [JETP Letters 108 (5), 312-317 (2018)] will be provided along with the Ref[1-14]. 

Authors should comment why the networks with non-adhesive and adhesive fillers are considered. Is there any other type of netwroks. The choose of orientation of fillers also should be commented for general reader. 

The explanation should be added in manuscript - what is the physical meaning of different  heaating time for EP (300s) and  PF 120 s? 

Authors stated that EP reaches a high cross-link density compared to PF and stated that SrFeO is partly integrated in the matrix, whereas NdFeB is loose in the network structure without adhesive forces. What is the physical origin of this 

and could it be sustitude, i.e.  NdFeB is integrated , whereas SrFeO is loose 

?

Table1 should be edited. (letters are overlapped)

Author Response

(The authors gave the same response as above.)
